# On the Loss of Context Awareness in General Instruction Fine-tuning

**Yihan Wang***
UCLA
wangyihan617@gmail.com

**Andrew Bai***
UCLA
andrewbai@ucla.edu

**Nanyun Peng**
UCLA
violetpeng@cs.ucla.edu

**Cho-Jui Hsieh**
UCLA
chohsieh@cs.ucla.edu

## Abstract

Pre-trained Large Language Models (LLMs) require post-training methods such as supervised fine-tuning (SFT) on instruction-response pairs to enable instruction following. However, this process can cause forgetting in capabilities learned during pre-training. In this paper, we investigate the loss of context awareness after SFT, where context awareness is defined as the ability to extract and understand information from user-provided context. Surprisingly, we discovered that the loss of context awareness occurs in instruction fine-tuned LLMs when the chat template is applied to input prompts. We identify that the performance decline is associated with a bias toward different roles learned during conversational instruction fine-tuning. The bias can be traced to training samples where the assistant response minimally relies on the user-provided instruction. Based on these observations, we propose a metric to identify context-dependent examples from general instruction fine-tuning datasets. We then apply conditional instruction fine-tuning with a context-dependency indicator, enabling the model to preserve context awareness after SFT. Experiments on four context-dependent downstream tasks and three pre-trained LLMs of different sizes show that our method effectively mitigates the loss of context awareness without compromising general instruction-following capabilities.

## 1 Introduction

Large language models (LLMs) pretrained on large-scale datasets acquire diverse language skills during pretraining. To enhance these models' ability to follow general instructions, further fine-tuning is typically required. This includes supervised instruction fine-tuning (SFT) [23, 20] and reinforcement learning from human feedback (RLHF) [6]. However, several studies have demonstrated additional fine-tuning can potentially harm existing capabilities learned during pretraining [17, 3, 10].

Although some studies suggest that performance degradation can be mitigated or even eliminated through improved instruction fine-tuning methods [19, 3, 11], in this paper, we demonstrate that instruction fine-tuning specifically leads to the worsening of a model's context awareness in a series of open-source models. We define context awareness as a model's ability to accurately retrieve, process, and interpret specific information from user-provided context. Context awareness is highly

---

*Equal contribution
Code is available at `https://github.com/YihanWang617/context_awareness`.

relevant to the *intrinsic hallucinations* of LLMs [13] and crucial to the truthfulness of LLM-based chat models [20]. It is also important for many real-world use cases, including retrieval augmented generalization [15, 14, 26], in-context learning [1], and contextual question-answering [21, 5, 8].

We first demonstrate the loss of context awareness through evaluations of several popular, open-source LLMs using the Needle-in-a-Haystack (NIH) task. We show that while many pretrained models demonstrate near-perfect performance on NIH, their performance deteriorates consistently after SFT, regardless of context window sizes, chat templates, architectures, or model sizes. We show that this decline is correlated with the application of chat templates, which, however, are widely used and essential in building conversational LLM assistants. When these chat templates are removed from instruction fine-tuned models, NIH performance not only recovers but in some instances surpasses that of their pretrained, non-fine-tuned counterparts.

These observations suggest that the deterioration in NIH performance does not indicate a catastrophic loss of context-retrieval capabilities during instruction fine-tuning. Instead, the chat template appears to mask these underlying abilities by introducing systematic biases into the models' behavior. Through analysis, we observe differences in attention allocation patterns in input tokens when comparing instruction fine-tuned models with and without applying chat templates. Specifically, we examine how the attention allocation shifts when tokens are marked as "user tokens" by the chat template. As illustrated in Figure 3, the application of chat templates leads to a redistribution of attention values: attention scores decrease for user input tokens while increasing for assistant tokens. We further validate the relationship between attention score reallocation and context awareness through targeted intervention experiments. By manually increasing attention values assigned to user tokens, we partially restored the performance on simple context-relevant tasks.

These findings motivate us to develop a fine-tuning strategy that mitigates attention bias acquired during instruction fine-tuning. Our approach stems from the intuition that the bias is learned from certain patterns in the training dataset, where for some examples the model does not need the context to generate correct answers. Therefore, we develop a quantitative metric to assess the context-dependency of conversational instruction and response pairs based on attention allocation patterns. We discover that context-dependent examples are notably sparse in commonly used open-source instruction fine-tuning datasets. To help the model distinguish examples with and without context-dependency during instruction finetuning, we add an indicator token to the identified context-dependent instructions. After fine-tuning the model with this enhanced dataset, it learns to allocate increased attention to user tokens when the indicator is present.

We evaluate our method on three open-source pretrained language models and several context-dependent and general tasks. Empirical results demonstrate that models fine-tuned using our method consistently achieve superior performance on context-dependent tasks compared to standard fine-tuning while maintaining similar performance on general tasks.

Our contributions are summarized as follows:

- We identify that the context awareness of LLMs deteriorates after supervised instruction fine-tuning with chat templates applied, compared to pretrained models.

- We pinpoint that the worsened context awareness is associated with attention allocation bias on tokens marked as from different roles.

- We propose a quantitative metric to identify context-dependent instruction-response pairs from general instruction fine-tuning datasets. By inserting an indicator into identified instructions during SFT, we mitigate the loss of context awareness in instruct models when the indicator is added during inference, while preserving general performance.

## 2   Loss of Context Awareness after Instruction Fine-tuning

We conduct preliminary studies to understand the loss of context awareness after instruction fine-tuning and its root cause. In Section 2.1, we present evidence that context awareness consistently deteriorates in instruct models and identified the main culprit as the roles indicated with chat templates. In Section 2.2, we analyze correlational relation between instruct models and decreased attention allocated to user role tokens. In Section 2.3, we find causal relation between the decreased attention allocated to user role tokens (after instruction fine-tuning) and worsened context awareness.

## 2.1 Evaluating Context Awareness Through Needle-in-a-Haystack (NIH) Testing

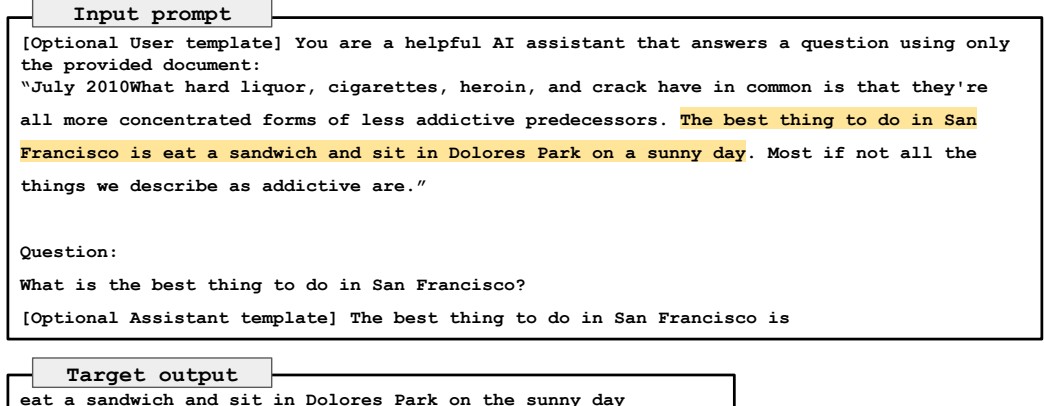

Figure 1: An example of the Needle-in-a-haystack (NIH) test used in our work. [Optional User template] and [Optional Assistant template] are user and assistant role indicators used in instruction fine-tuned models. The inserted needle is highlighted in yellow.

We demonstrate the loss of context awareness after instruction fine-tuning with the needle-in-a-haystack (NIH) test. The NIH task provides a fairer comparison between pretrained models and instruct models in terms of context retrieval performance, since it relies less on instruction-following capabilities. We remove the newlines between context and needle in the original NIH test to increase difficulty and better discriminate among different models. An example of the NIH prompt is shown in Figure 1. We rerun the evaluation with different prompt templates for a more robust evaluation. More details can be found in Appendix A.2.

**Dataset.** The NIH test evaluates the performance of language models in extracting a given sentence (the needle) from irrelevant context. The needle can be inserted at different locations in contexts of varying lengths. We report the recall error:

$$\text{recall} = \frac{1}{|K|} \sum_{w \in K} \mathbb{1}(w \in output)$$

$$\text{err} = 1 - \text{recall}$$

where $K$ is the set of keywords in the targeted output and $output$ is the output of the LLM. For all NIH evaluations, we calculate the recall on the first 100 generated tokens. We average the recall error across 400 NIH tests with different insertion locations and context lengths within the model's context window. An example of the NIH prompt in our experiments is shown in Figure 1. When the chat template is applied to the prompt, the whole input prompt is partitioned into the user instruction input and model response, indicated by special role markers in the chat template (e.g., <|user|> and <|assistant|>). More details about the NIH tests can be found in Appendix A.2.

**Models.** We evaluate NIH on eight open-source language models from five model families. For each model, we compare the performance of the pretrained version (not instruction fine-tuned) and the official instruction-finetuned version released by the model provider. Here, we do not consider stronger closed-source models as their pretrained versions are unavailable. The context window lengths of these models range from $4K$ to $32K$.

**NIH performance drops after instruction finetuning on most models.** We report the evaluation results on NIH in Figure 2. Given the significantly improved instruction-following through instruction finetuning, we would expect that performance would **always** increase. However, when comparing the pretrained model (green bar) with the instruction-finetuned model (red bar), the NIH error *increases* for most models after instruction fine-tuning, which implies negative effects from worse context awareness after finetuning. The only outlier is Llama-3.1-8b, which highlights the nuanced dual impact of instruction fine-tuning on different models: improvement in instruction following and potential worsening of context sensitivity.

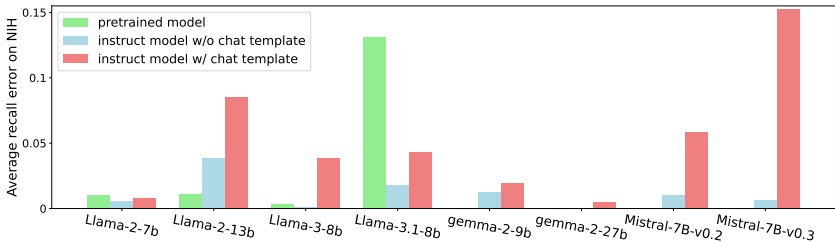

Figure 2: Average recall error (1 - recall) on NIH for different model series (lower better). We report the performance of official instruction-tuned models (both with and without chat templates) and their pretrained counterparts from five model families, with sizes ranging from 7B to 27B. Some errors are too small to be visible in the figure. Detailed numerical values can be found in Appendix B.1.

**The performance drop is associated with chat templates.** To determine whether the performance difference mainly comes from a different input format or fine-tuned model weights, we remove the chat templates (i.e., the role indicators in instruction-tuned models) and visualize the NIH errors with blue bars in Figure 2. The NIH error without applying chat templates (blue bar) is significantly lower than with templates (red bar). These results indicate that context retrieval capabilities are not eliminated by instruction fine-tuning, but are instead impacted by biases associated with the presence of chat templates.

The aforementioned phenomenon is consistent across models with varying context window lengths, model families, chat templates, and small to medium model sizes. Although we are unable to conduct experiments on extremely large models, context awareness in medium-sized models remains relevant, as they are widely adopted in cost-sensitive settings such as edge devices and small businesses.

## 2.2 Attention Allocation Bias Across Different Roles

Based on our observations, performance on NIH drops significantly when the chat template is applied. We hypothesize that the performance deterioration stems from the bias in instruction data and the bias is embedded in different roles marked by the chat template. When the model generates a response, it balances information from the input context and internal knowledge stored within its weights. It pays attention to user tokens to maintain consistency with the user-provided context while attending to previously generated assistant response tokens to maintain consistency with its output. If the model learns to assign lower importance to user-provided context and higher importance to its internal knowledge during SFT, it may develop a bias that causes it to weigh user tokens less. To support our hypothesis, we analyze the attention allocation between user and assistant tokens, both with and without chat templates.

**Experimental settings.** We prepare two inputs for each NIH test case: one with the chat template and one without. The prompt formats follow the input prompt shown in Figure 1. We collect attention weights from each layer, focusing on the last token (which generates the next answer token) and its attention to all input tokens. We separately sum the attention weights for user and assistant tokens. When calculating the attention weight allocation with chat templates, we exclude the attention weights on chat template tokens and renormalize the attention weights across the user, assistant, and BOS tokens. We report the attention allocation from an arbitrary middle layer (e.g., Layer 15) on a representative context retrieval head that allocates the highest attention to user tokens without chat templates. Further discussion on head and layer selection can be found in the Appendix B.3.

**Less attention on user tokens with chat templates.** We visualize the changes in attention allocation, both with and without chat templates in Figure 3. When chat templates are applied to mark tokens as from different roles, attention allocated to user tokens decreases while attention to assistant tokens increases for all models. This indicates that the models learn to assign lower attention to user tokens compared to the baseline level (where the chat template is not applied). In our experiments, we collect attention allocation data for context lengths less than 4,000. Although several models (e.g. Llama-3 and Gemma-2) achieve perfect NIH accuracy under 4,000 context length, the decrease in

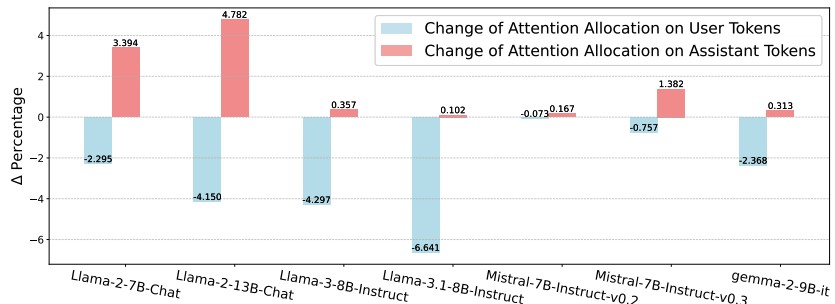

Figure 3: We visualize the changes in attention allocation between user tokens and assistant tokens after applying chat templates. The attention allocation is calculated when the model generates the first answer token in its response. The attention weights are averaged across 400 tests with context lengths ranging from 200 to 4,000 and needle depths from 0% to 100%. More detailed scores can be found in Appendix B.2.

user attention remains noticeable. Note that comparison is only reasonable within variants of each model and not between different ones.

## 2.3 Attention Steering to Compensate for Attention Bias

In the previous section, we observed a trend of decreased attention allocated to user tokens when chat templates were applied to instructional models, associated with a performance decline on the NIH task. To establish a more robust causal relationship, we further verify our hypothesis by manually steering attention toward user tokens to compensate for the attention bias.

**Post-hoc attention steering of user tokens.** To compensate for the attention bias observed in instruction models, we manually steer the attention on user tokens.

Specifically, we modify the self-attention weights in each transformer layer:

$$\hat{\text{Att}}(\mathbf{x}, \mathbf{y}) = \begin{cases} \frac{1}{Z} \cdot \alpha \text{Att}(\mathbf{x}, \mathbf{y}) & \text{if } \mathbf{y} \notin U \\ \frac{1}{Z} \cdot \text{Att}(\mathbf{x}, \mathbf{y}) & \text{otherwise,} \end{cases} \tag{1}$$

where $\mathbf{x}$ and $\mathbf{y}$ are two tokens in the input sequence, $\alpha \in (0, 1)$ is the steering strength (lower for more emphasis on user tokens), $U$ is the subset of all user tokens, and $Z$ is the normalization constant that renormalizes the altered attention scores across all tokens. $\text{Att}(\mathbf{x}, \mathbf{y})$ is the original attention weight from token $\mathbf{x}$ to token $\mathbf{y}$.

We adopt the same attention steering implementation as Zhang et al. [28]. They steered the attention of pretrained language models to emphasize user-specified portions of user instructions, enabling models to follow instructions without explicit instruction fine-tuning. In our setting, we increase the attention weights of instruction-fine-tuned models on the entire user input prompt, which consequently decreases weights on other tokens (chat template role tokens, BOS/EOS tokens, and partially generated model responses). We steer on all heads with intervention factor $\alpha = 0.95$.

**Post-hoc attention steering partially recovers the NIH performance but produces side effects.** We report the performance of attention steering in Table 1. Attention steering requires customized attention calculations for different heads and layers, which limits the use of several existing efficient attention implementations. Therefore, we are only able to apply attention steering to two Llama-2 models with 4,000-token context windows. We report the recall on NIH task as well as the performances on two additional contextual QA tasks: QuAC [5] and DROP [8]. Detailed descriptions and metrics of these tasks can be found in Section 4. Unlike NIH and QuAC, which retrieve exact sentences from the context, DROP requires the model not only to understand and retrieve relevant information from the context but also to apply discrete mathematical operations to the retrieved information. As shown in the table, attention steering can boost performance on simple context retrieval tasks such as NIH and QuAC. However, on DROP, which requires a more complex combination of different capabilities, performance with attention steering is negatively impacted. The

Table 1: NIH recall and QuAC/DROP containing score with attention steering. "Baseline" and "+ Attention Steering" are evaluated with chat templates. "w/o chat template" shows the NIH performance without the chat template for reference (same as Figure 2).

| Task | Capabilities | Model Name | Baseline | + Attention Steering | w/o chat template |
|------|-------------|------------|----------|---------------------|-------------------|
| NIH | sentence retrieval | Llama-2-7B-Chat | 0.9917 | 0.9932 | 0.9975 |
| | | Llama-2-13B-Chat | 0.9207 | 0.9225 | 0.9578 |
| QuAC | sentence retrieval reading comprehension | Llama-2-7B-Chat | 22.20 | 24.00 | - |
| | | Llama-2-13B-Chat | 18.60 | 20.00 | - |
| DROP | context retrieval math operation | Llama-2-7B-Chat | 44.22 | 43.46 | - |
| | | Llama-2-13B-Chat | 46.20 | 45.11 | - |

deteriorating performance on DROP suggests that intervening in attention scores to emphasize user tokens, while improving context awareness, might impair other capabilities of the model.

In the next section, we introduce a fine-tuning strategy to better mitigate the loss of context awareness with fewer side effects based on all of our aforementioned observations.

## 3  Instruction Fine-tuning with Context-dependency Indicators

Our method is based on the intuition that by explicitly marking context-dependent data samples with a special indicator during instruction fine-tuning, the model learns to associate the indicator with paying more attention to the user-provided context. After fine-tuning whenever the indicator is appended to a user instruction, the conditional generation allocates more attention to the user-provided content and responds to the instruction with more context awareness. The main technical challenge is identifying context-dependent data samples from the instruction dataset.

### 3.1  Identifying Context-dependent Instructions

A training sample in the instruction fine-tuning dataset is a conversation between the user and model assistant, which may consist of multiple instruction-response pairs. Formally, we denote user instructions as $X$, assistant responses as $Y$, and a conversation of $n$ total turns as $C = [X_1, Y_1, \ldots, X_n, Y_n]$.

**Identifying context-dependent instructions with a reference language model.** We identify the context-dependent instructions by calculating the attention allocation on user tokens. We start by preparing a seed instruction fine-tuned model $M$, which can be the same or a weaker pretrained model fine-tuned with the original instruction fine-tuning dataset. We then define the context-dependency score for the $m^{\text{th}}$ turn response $Y_m$ given its instruction $\mathbf{X}_m$ and conversation history:

$$s_M(\mathbf{Y}_m) = \frac{1}{|\mathbf{Y}_m|} \sum_{\mathbf{y} \in \mathbf{Y}_m} \max_{h \in H} \left( \sum_{\mathbf{x} \in \mathbf{X}_1 \cup \ldots \cup \mathbf{X}_m} \text{Att}_h(\mathbf{y}, \mathbf{x}) \right), \tag{2}$$

where $H$ is the set of attention heads in model $M$ and $\text{Att}(\mathbf{y}, \mathbf{x})$ is the attention weight from token $\mathbf{y}$ to $\mathbf{x}$. Intuitively, the score $s_M(\mathbf{Y}_m)$ measures the sum of attention scores allocated to all user instructions in prior turns $\mathbf{X}_1 \cup \ldots \cup \mathbf{X}_m$, averaged over response tokens $\mathbf{y} \in \mathbf{Y}_m$. As different heads learn different capabilities, we calculate the score on the most representative head for context retrieval on each layer, specifically the head that allocates the highest attention weight to user tokens. We compute the score on a single middle layer for practical efficiency, as we find the relative scores $s_M$ to be insensitive to layer choice. We defer the detailed discussion of layer and head selection to Appendix B.5.

### 3.2  Instruction Fine-tuning with Context-dependency Indicators

A threshold $\beta \in (0, 1)$ can be selected after the context-dependency score is obtained for each instruction-response pair. A conversation turn $(\mathbf{X}_m, \mathbf{Y}_m)$ with $s_M(\mathbf{Y}_m) > \beta$ is considered context-dependent. We append a special token [IND] to the user instruction $\mathbf{X}_m$ if it is context-dependent. In our implementation, the special token [IND] is added as an additional special token to the vocabulary to avoid conflicts with existing ones.

After conditional instruction fine-tuning, the user can specify whether to add this indicator to their query, depending on whether the model response should rely more on user-provided context.

# 4 Experiments

We validate the effectiveness of our method using three open-source pretrained models trained on three instruction fine-tuning datasets and benchmarked a set of context-dependent and general tasks.

## 4.1 Experiment Settings

**Models** We evaluate our method on three open-source pretrained large language models: TinyLlama-1.1B [27], Llama-2-7B [22], and Llama-3-8B [9]. TinyLlama-1.1B is a 1.1B Llama model pretrained on 3 trillion tokens with a context window length of 2048. Llama-2-7B and Llama-3-8B have context windows of 4096 and 8192 tokens, respectively. Due to limited computational resources in academic labs, we can only fine-tune models with up to 8B parameters. We also truncate the training examples to 4096 tokens. Detailed hyperparameters can be found in Appendix A.1.1.

**Instruction Fine-tuning Datasets** We experiment with three popular open-source instruction fine-tuning datasets: ShareGPT, adopted by Vicuna [4], UltraChat-200k [7], and WizardLM-70K [25]. For ShareGPT, we follow the same preprocessing process as Chiang et al. [4]. We remove refusal responses from ShareGPT and WizardLM-70K to prevent the fine-tuned models from becoming oversensitive and frequently refusing to respond. For all three datasets, we remove model responses from incomplete conversation chunks that lack user input instructions. Statistics of the processed datasets are presented in Table 2.

Table 2: Statistics of instruction fine-tuning datasets in our experiments. We report the statistics after performing preprocessing as detailed in Section 4.1. Average length is measured in the number of tokens with TinyLlama tokenization.

| Datasets | Avg. conversation length | # conversations | # instructions |
|---|---|---|---|
| ShareGPT | 1,567.68 | 93,645 | 331,722 |
| UltraChat-200k | 1,437.33 | 207,865 | 657,794 |
| WizardLM-70K | 484.00 | 57,523 | 57,523 |

**Context awareness benchmarks.** In addition to NIH, we report the performance on three closed-book QA tasks to benchmark context awareness: SQuAD [21], QuAC [5], and DROP [8]. SQuAD is a reading comprehension benchmark where the answer to each question can be found in the context. We evaluate only the answerable subset of questions in SQuAD 1.0. QuAC is similar to SQuAD, but its questions are more open-ended and the lengths of the answers are longer. While NIH, SQuAD, and QuAC only require direct retrieval from context, DROP requires more complicated reasoning based on the given context, and its answers require discrete operations on the retrieved context such as addition, sorting, or counting.

As instruction fine-tuned models are not specifically trained on QA tasks to provide concise answers, their responses are generally more verbose. Therefore, we report the containment score, defined as whether the model response contains the ground-truth answer with keyword string matching, rather than the F1 score to exclude the effects of different models' response styles. Prompt templates for QA tasks are listed in Appendix A.3.

For Needle-in-a-haystack, we report the recall defined in Section 2.1, which is also the default metric used in Dubey et al. [9]. We set the maximum NIH context length to 1,000 for models fine-tuned on WizardLM-70K due to its shorter instruction lengths. For models fine-tuned on ShareGPT and UltraChat-200K, we set the maximum NIH context length to the maximum context window considered in fine-tuning, which is 2,000 for TinyLlama and 4,000 for Llama-2 / Llama-3. The prompt template used in NIH is the same as Section 2.1 except that we remove the response prefix and keep only the user input prompt.

**General instruction-following benchmarks.** To validate that our method maintains strong performance on general instruction-following tasks, we evaluate the fine-tuned models on MT-Bench

[29] where the response quality is rated by a GPT-4 judge based on helpfulness, relevance, accuracy, depth, creativity, and level of detail. We report the average rating across the MT-Bench test cases.

Table 3: Comparing vanilla instruction finetuning with finetuning with context-relevant indicators (+ indicator). For "+ Indicator" models, `[IND]` is added in all evaluations. As a reference, we also list the performances evaluated on official Llama-2 and Llama-3 instruct models, which are finetuned with closed-source datasets. NIH, SQuAD, and QuAC are simple context-dependent tasks, while DROP and MT-Bench require more complex capabilities.

| SFT dataset | Pretrained Model | Method | Context-dependent tasks | | | Complex-skill tasks | |
|---|---|---|---|---|---|---|---|
| | | | NIH | SQuAD | QuAC | DROP | MT-Bench |
| ShareGPT (Vicuna) | TinyLlama-1.1B | Vanilla | 0.9846 | 59.73 | 15.50 | 27.39 | 3.7250 |
| | | + Indicator | **0.9921** | **62.05** | **17.40** | **27.84** | **3.7375** |
| | Llama-2-7b | Vanilla | 0.3378 | 76.78 | 23.60 | 33.90 | **6.4875** |
| | | + Indicator | **0.7007** | **79.09** | **24.20** | 33.90 | 5.7375 |
| | Llama-3-8b | Vanilla | 0.8957 | 83.06 | **24.80** | 42.15 | **7.4375** |
| | | + Indicator | **0.9404** | **84.80** | 24.50 | **43.17** | 7.1625 |
| UltraChat-200K | TinyLlama-1.1B | Vanilla | 1.0000 | 73.03 | 22.70 | 30.96 | 3.9000 |
| | | + Indicator | 1.0000 | **74.47** | **23.10** | 30.96 | **4.1125** |
| | Llama-2-7b | Vanilla | **0.9850** | 83.81 | 24.20 | **37.91** | 5.7125 |
| | | + Indicator | 0.9725 | **85.76** | **26.10** | 37.58 | **5.8125** |
| | Llama-3-8b | Vanilla | 1.0000 | 85.12 | 25.50 | **50.99** | **7.2375** |
| | | + Indicator | 1.0000 | **86.28** | **26.40** | 50.22 | 6.8500 |
| WizardLM-70K | TinyLlama-1.1B | Vanilla | 0.9250 | 60.51 | 13.80 | 27.53 | 4.2750 |
| | | + Indicator | **0.9925** | **63.39** | **14.60** | **28.36** | **4.3000** |
| | Llama-2-7b | Vanilla | 0.7375 | 82.89 | 23.70 | 34.07 | 5.7750 |
| | | + Indicator | **0.9254** | **83.13** | **25.30** | **34.44** | **6.2250** |
| | Llama-3-8b | Vanilla | 0.9846 | 88.25 | 24.60 | 46.87 | 7.1125 |
| | | + Indicator | **0.9871** | **88.53** | **26.00** | **47.85** | **7.5250** |
| (Closed-source datasets) | Llama-2-7b-chat | - | 0.8264[*] | 83.28 | 22.20 | 44.22 | 6.9375 |
| | Llama-3-8b-Instruct | - | 1.0[*] | 86.96 | 27.40 | 46.54 | 8.0750 |

[*] Here NIH is evaluated without the response prefix used in Section 2.1 and the maximum context length is set to 4096 for fair comparison. Therefore, the exact numbers differ from Figure 2.

## 4.2 Instruction Fine-tuning with Context-dependency Indicators

**Settings and hyperparameters.** We adopt a TinyLlama model fine-tuned on the original ShareGPT (Vicuna) dataset as the seed model $M$ and compute the context-dependency score on a middle layer (15 in all of our main experiments) for faster computation. We set the threshold for context-awareness as $\beta = 0.6$ for all experiments reported in Table 3. An ablation study on the choice of threshold value can be found in Appendix B.6.

**Sparsity of context-dependent instructions.** We compute the context-dependency scores on all three instruction fine-tuning datasets (see Table 10) and find that context-dependent instructions are consistently scarce in all datasets. Note that the scarcity is an intrinsic property of the datasets overlooked by the original curators of the data. This observation supports our hypothesis that the model learns the bias to weigh user tokens less importantly from the instruction dataset.

## 4.3 Experiment Results

**Conditional finetuning improves performance on context-dependent tasks.** Table 3 shows that "+ Indicator" (ours) outperforms "vanilla" fine-tuning consistently across different models and SFT datasets on NIH, SQuAD, and QuAC. The benchmarks isolate and measure context awareness performance. For the "vanilla" fine-tuning setting, we train and evaluate the model without the indicator token. For the "+ Indicator" setting, we add the indicator token to the selected subset of prompts in fine-tuning and all queries for evaluation. The results confirmed that models learn to focus more on the user-provided context when the indicator token is present in the prompt.

**Sparsity of context-dependent instructions impacts performance on general tasks.** The purpose of evaluating on general tasks is to demonstrate that our method minimally impacts other capabilities than context awareness. Table 3 presents the evaluation results on MT-Bench as an assessment of general instruction-following capabilities and DROP to assess more complex reasoning capabilities inadditionl to purely contextual retrieval. Models fine-tuned with the indicator perform comparably or sometimes better. Comparing the performance between the three SFT datasets, ShareGPT suffers from the most negative impact with the indicator, while WizardLM-70K improves. This can be explained by the net number and thus diversity of the context-dependent subset in each dataset. Table 10 shows that WizardLM-70K has the highest proportion of context-dependent samples, while ShareGPT has the least.

## 5 Related Work

**Instruction fine-tuning and chat templates.** To enable instruction-following, pretrained LLMs usually require supervised fine-tuning on instruction-following datasets (SFT) [23, 20], and optionally reinforcement learning with human feedback (RLHF) [6]. Instruction fine-tuning datasets consist of user instruction and target model response pairs, which can be collected from modified NLP tasks [23, 19], human annotations [20, 4] or synthesized data from existing LLMs [7, 25]. Instruction fine-tuning usually converts training examples into a dialog format with a chat template, which typically consists of user, assistant, and system role indicators. However, these role indicators and role partition in the conversation are not sufficiently presented and learned during pretraining, making them prone to bias during fine-tuning.

**Context awareness and hallucinations in LLMs** Context awareness is crucial for mitigating hallucinations where the response is not consistent with the provided context, such as the "closed-domain" hallucination in Ouyang et al. [20] and intrinsic hallucination in Huang et al. [13]. Several existing works aim to understand and mitigate these intrinsic hallucinations. Liu et al. [18] studies the failure of context retrieval when the relevant information is in the middle of the provided context, showing the existence of positional bias in context retrieval. Follow-up work [12] proposes to calibrate this positional bias and mitigate the issue. In our work, we study a new **role bias** in popular open-source instruction-finetuned models, where the context receives less attention when marked as user tokens by the chat template. Some other papers such as An et al. [2] synthesize examples targeted toward specific tasks (e.g., contextual QA) to increase the performance on context-dependent tasks. Instead, our method is more general in terms of task and input format compared. This makes our method more applicable to different types of user queries that require greater attention to user-provided context.

**Side effects of instruction finetuning** Neural networks are known to catastrophically forget existing knowledge or capabilities when sequentially trained on new tasks or domains [16]. It is commonly believed that traditional catastrophic forgetting on pretraining-stage capabilities can be significantly mitigated by finetuning the model on a diversified mixture of prompts [19, 3, 11]. We discovered that, contrary to popular belief, context awareness can deteriorate after instruction fine-tuning.

## 6 Conclusion

This work highlights the detrimental effects of supervised instruction fine-tuning on the context awareness of pretrained language models, even in scenarios involving short context lengths. We have identified that the decline in context awareness is closely linked to attention allocation biases within chat templates, which are learned during conversational instruction finetuning. Our proposed method utilizes conditional supervised fine-tuning with an indicator marking samples with context-relevant training samples. Our method effectively maintains contextual understanding while benefiting from supervised instruction.

**Limitation** Due to computational resource limitations, our methods were only validated on smaller models. We hope to extend the experiments to full finetuning or larger models when more resources become available. Additionally, our technique of associating context-dependent user instructions with the indicator token may also encode other unintended styles from the selected subset of instructions. This issue is particularly aggravated when the subset of context-relevant samples is small and

significantly different from the remaining dataset. One future direction is to better disentangle the context-dependency signal within the selected corpus. Another future direction is expanding the evaluation of context-dependent benchmarks (e.g., RAG) to discover the extent of the harm caused by the decreased context awareness.

## Acknowledgments and Disclosure of Funding

This work is partially supported by NSF 2048280, 2325121, 2244760, 2331966 and ONR N00014-23-1-2300:P00001.

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

# A  Appendix

## A.1  Experimental Details

### A.1.1  Instruction Fine-tuning

We adopted the fine-tuning recipes from the Huggingface alignment-handbook[2] for Llama-2 and Llama-3 QLoRA tuning. For the TinyLlama model, we used the fine-tuning recipe provided by the author[3]. See detail configurations in Table 4. We fine-tuned the models for one epoch on ShareGPT and UltraChat-200K, and two epochs on WizardLM-70K due to its smaller training set. We used the TinyLlama chat template for all instruct models fine-tuned in Table 3. All experiments are conducted on 4 A6000 GPUs on a local server. Execution time of training runs usually range from a couple hours to at most 1-2 days.

Table 4: Fine-tuning hyperparameters configuration

| Models | Fine-tune config | Learning rate | Batch size | Precision |
|---|---|---|---|---|
| TinyLlama | Full fine-tune | 2e-5 | 128 | bf16 |
| Llama-2/3 | QLoRA with rank = 16, alpha =16 | 2e-4 | 64 | bf16 |

## A.2  NIH Evaluation Details

For all NIH evaluations, we average the recall error across 400 tests. Specifically, we evaluate on 20 context lengths uniformly distributed between 200 and the maximum context length, and 20 needle insertion depths uniformly located between 0% and 100%.

To ensure that our NIH evaluation is not sensitive to differences in prompt templates, we run evaluations with 4 different prompt templates on small-scale experiments and report the mean and standard deviation. For evaluations on larger models or larger context window sizes, we report the evaluation results using only one prompt template. We illustrate the mean errors in Figure 1. Complete results with standard deviations can be found in Table 7.

## A.3  Contextual QA Evaluation Details

We list the prompts used in contextual QA tasks in Table 5 and Table 6. For contextual QA tasks, we generate answers of up to 100 tokens and truncate them at the end of the first complete sentence. For NIH tests, we generate answers of up to 50 tokens.

As UltraChat-200K constructs its data with a fixed set of prompt templates similar to our default ones used in evaluation (the templates used for ShareGPT and WizardLM models in Table 6 and 5), we evaluate UltraChat-200K-finetuned models with a simpler template to exclude the impact of overfitting on finetuning prompt templates.

Table 5: Prompt templates used for SQuAD and DROP in Table 1 and Table 3 when the model is finetuned on different instruction finetuning datasets.

| Instruct Finetuning Dataset | Template for SQuAD and DROP |
|---|---|
| ShareGPT & WizardLM-70K | {context}\nAnswer the question according to the above passage: {question} |
| UltraChat-200K | {context} {question} |

---

[2] https://github.com/huggingface/alignment-handbook
[3] https://github.com/jzhang38/TinyLlama

Table 6: Prompt templates used for QuAC in Table 1 and Table 3 when the model is finetuned on different instruction finetuning datasets.

| Instruct Finetuning Dataset | Template for QuAC |
|---|---|
| ShareGPT & WizardLM-70K | {context}\nAnswer the question with pieces from the the above passage: {question} |
| UltraChat-200K | {context} {question} |

# B Additional Experiment Results

## B.1 Full NIH Results on Open-source Official Models

In Figure 2, we only report the NIH performances when the response prefix is added, for fair comparison. In Table 7, we show the exact numbers for Figure 2 as well as additional evaluation results without the response prefix. When the response prefix is removed, the performance drop on NIH is even more significant compared to results without chat templates.

The standard deviations shown in the table are explained in Section A.2.

Table 7: NIH performance with and without chat templates on different models. The mean and standard deviations were calculated using 4 different prompt templates listed in Table 8.

| Model Name | Context window | w/o chat template w/ response prefix | w/ chat template w/ response prefix | w/ chat template w/o response prefix |
|---|---|---|---|---|
| Llama-2-7b | 4K | $98.94 \pm 0.33\%$ | - | - |
| Llama-2-7b-chat | | $99.40 \pm 0.48\%$ | $99.17 \pm 0.28\%$ | $92.45 \pm 1.24\%$ |
| Llama-2-13b | 4K | $98.89 \pm 0.34\%$ | - | - |
| Llama-2-13b-chat | | $96.13 \pm 0.64\%$ | $91.50 \pm 0.76\%$ | $92.78 \pm 0.24\%$ |
| Llama-3-8b | 8K | $99.62 \pm 0.26\%$ | - | - |
| Llama-3-8b-instruct | | $99.92 \pm 0.09\%$ | $96.10 \pm 0.58\%$ | $95.89 \pm 0.62\%$ |
| Llama-3.1-8b | 128K | $86.89\%$ | - | - |
| Llama-3.1-8b-instruct | | $98.17\%$ | $95.64\%$ | $94.75\%$ |
| mistral-v0.2 | 32K | $100\%$ | - | - |
| mistral-v0.2-instruct | | $99.00\%$ | $94.14\%$ | $93.92\%$ |
| mistral-v0.3 | 32K | $100\%$ | - | - |
| mistral-v0.3-instruct | | $99.32\%$ | $84.71\%$ | $72.00\%$ |
| gemma-2-9b | 8K | $100 \pm 0\%$ | - | - |
| gemma-2-9b-it | | $98.75 \pm 0\%$ | $98.03 \pm 0\%$ | $98.72 \pm 0.54\%$ |
| gemma-2-27b | 8K | $100\%$ | - | - |
| gemma-2-27b-it | | $100\%$ | $99.64\%$ | $99.25\%$ |

## B.2 Full Results for Figure 3

In Figure 3, we only show the changes in attention allocation with and without chat templates. In Figure 4, we show the absolute numbers of attention allocation for each part of the input prompts. When the chat template is added, we normalize the attention weights on user tokens, response tokens, and the BOS token only, with the sum of attention allocation being 1.

## B.3 Probing Context Retrieval Heads

In Sections 2.2, 2.3, and 3.1, we mentioned identifying a representative context retrieval head on each layer that allocates the largest attention to user tokens. Because different attention heads can have very different functionalities, and context retrieval heads can be sparse among all heads [24], we believe that selecting a representative context retrieval head for visualization, attention steering, and data selection is both necessary and important. Specifically, given an input sequence $\mathbf{x}_1, \mathbf{x}_2, \ldots, \mathbf{x}_m$ with $m$ tokens, we select the context retrieval head $h^*$ from each layer $l$ that allocates the highest

Table 8: Four different prompt templates were used in the NIH evaluation. In Table 7, we report the mean and standard deviation across different prompt templates for small models and small context windows. {context} represents the context with the needle inserted.

| Prompt templates in NIH evaluation |
| --- |
| You are a helpful AI assistant that answers a question using only the provided document: {context} Question: {retrieval_question} |
| You are a helpful AI assistant that answers a question using only the provided context: {context} Question: {retrieval_question} |
| Document: {context} Answer the question accoriding to the provided document: {retrieval_question} |
| Context: {context} Answer the question accoriding to the provided context: {retrieval_question} |

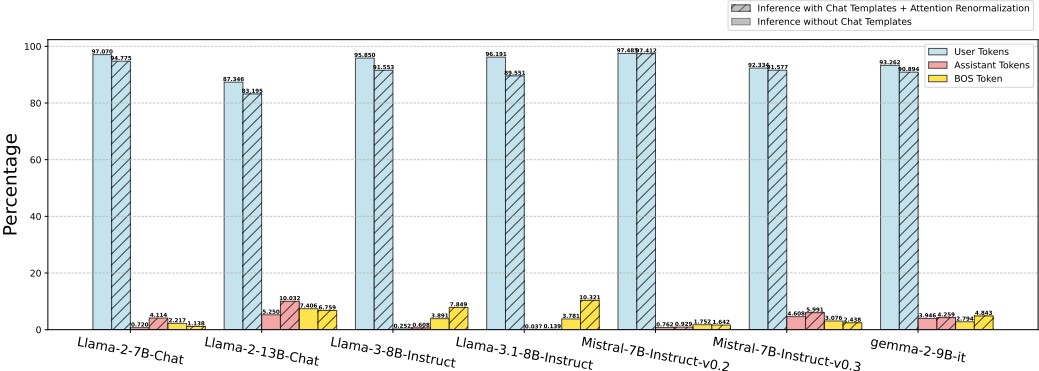

Figure 4: We visualize the full attention allocation on user tokens, assistant tokens, and BOS token with and without applying the chat templates. The attention allocation is calculated when the model is generating the first answer token in its response. For cases where the chat template is applied, we normalize the attention values on user tokens, assistant tokens, and the BOS token such that attention scores allocated to these three sum up to 1. The attention weights are averaged across 400 tests with context lengths ranging from 200 to 4000 and needle depths from 0% to 100%.

attention to user tokens when generating the first answer token:

$$h^* = \arg\max_{h \in H_l} \sum_{\mathbf{x}_i \in U} \text{Att}_{h,l}(\mathbf{x}_{-1}, \mathbf{x}_i) \tag{3}$$

, where $\mathbf{x}_{-1}$ and $\mathbf{x}_i$ are the last token and $i$th token in the sequence of tokens on each layer, respectively. $U$ is a subset of all user tokens.

In Section 2.2, we select the retrieval head on layer 15 for each input prompt and visualize the average attention change on the selected head across different prompts. In Section 2.3, we select one retrieval head on each layer using a sampled NIH prompt and steer all identified retrieval heads. In Section 3.1, we select the retrieval head on layer 15 for each input prompt and calculate the context-dependency score. In Section B.5, we provide a further discussion on the agreement between different layers.

## B.4 Attention Distribution Analysis

This section compares the attention distributions of inference-time attention steering and training-time conditional SFT to illustrate why the latter method is better for recovering context-awareness. We analyze the attention allocated to user versus assistant responses in Llama-2, fine-tuned on ShareGPT. The attention steering method is applied with $\alpha = 0.95$, while the conditional SFT model was trained using a context-relevance filtering threshold of $\beta = 0.6$.

Fig 5 presents the attention distribution averaged across all heads (left) and the distribution for the retrieval head specifically (right). We observe that while attention steering boosts user attention more than conditional SFT when averaged across all heads, the opposite is true for the retrieval head. This discrepancy highlights the coarse-grained nature of attention steering; it uniformly steers all heads, which can unnecessarily boost attention in ways that do not improve context-awareness. Conversely, the conditional SFT method learns to *selectively* increase user attention on the critical retrieval head, thus avoiding the unintended performance regressions associated with indiscriminate attention manipulation.

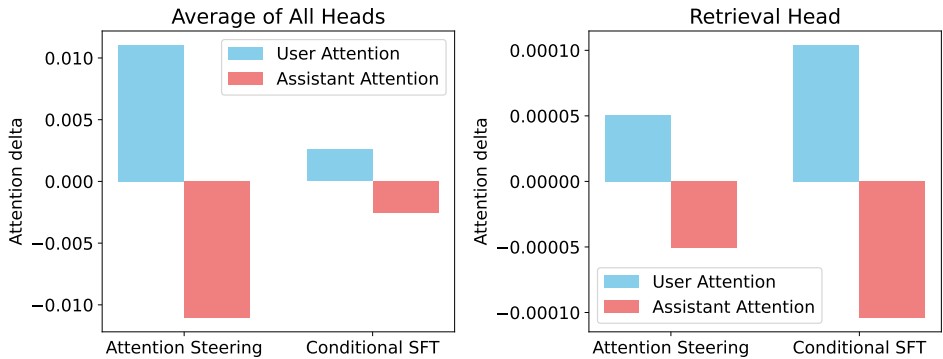

Figure 5: We visualize the attention distribution shift of the attention steering and conditional SFT variants relative to the vanilla instruction fine-tuned model.

## B.5  Agreement Between Different Layers

In Figure 6, we calculate and visualize the disagreement heatmap in $\hat{S}$ selection when the context-dependency score is calculated across different layers. We use the same TinyLlama model, fine-tuned on the vanilla ShareGPT dataset, as the seed model $M$. Specifically, we first calculate the context-dependency scores for each conversation turn in 500 randomly sampled examples from the ShareGPT dataset across different layers. We then select the top 10% of conversation turns with the highest context-dependency scores on each layer $l$ as the subset $\hat{S}_l$. We compute the disagreement between two layers $l$ and $l'$ by calculating the ratio of non-overlapping conversation turns in their respective subsets $\hat{S}_l$ and $\hat{S}_{l'}$. We can see from the figure that the disagreement among the 9 middle layers is low, indicating that we can safely choose an arbitrary layer for the context-dependency score calculation.

## B.6  Ablation Study for Different Threshold $\beta$

Table 9: Ablation study with different threshold $\beta$, which is used in Section 3.

| Threshold $\beta$ | SQuAD | QuAC | DROP | MT-Bench |
| --- | --- | --- | --- | --- |
| 1.0 (Vanilla) | 0.5918 | 0.1130 | 0.2739 | 3.725 |
| 0.5 | 0.6207 | 0.1270 | 0.2872 | 4.075 |
| 0.6 | 0.6144 | 0.1290 | 0.2784 | 3.825 |
| 0.7 | 0.6160 | 0.1290 | 0.2786 | 3.675 |

We use $\beta = 0.6$ in all our main experiments. To evaluate the sensitivity to the threshold $\beta$, we select $\hat{S}$ with different thresholds and prepare the final modified instruction finetuning dataset. We fine-tune a TinyLlama-1.1B model on these three datasets and evaluate it on three contextual QA tasks and MT-Bench. As shown in Table 9, all three models outperform vanilla finetuning on the contextual QA tasks. However, performance on MT-Bench shows a decreasing trend when the threshold increases from 0.5 to 0.7, potentially due to a more drastic difference between $\hat{S}$ and the unselected subset.

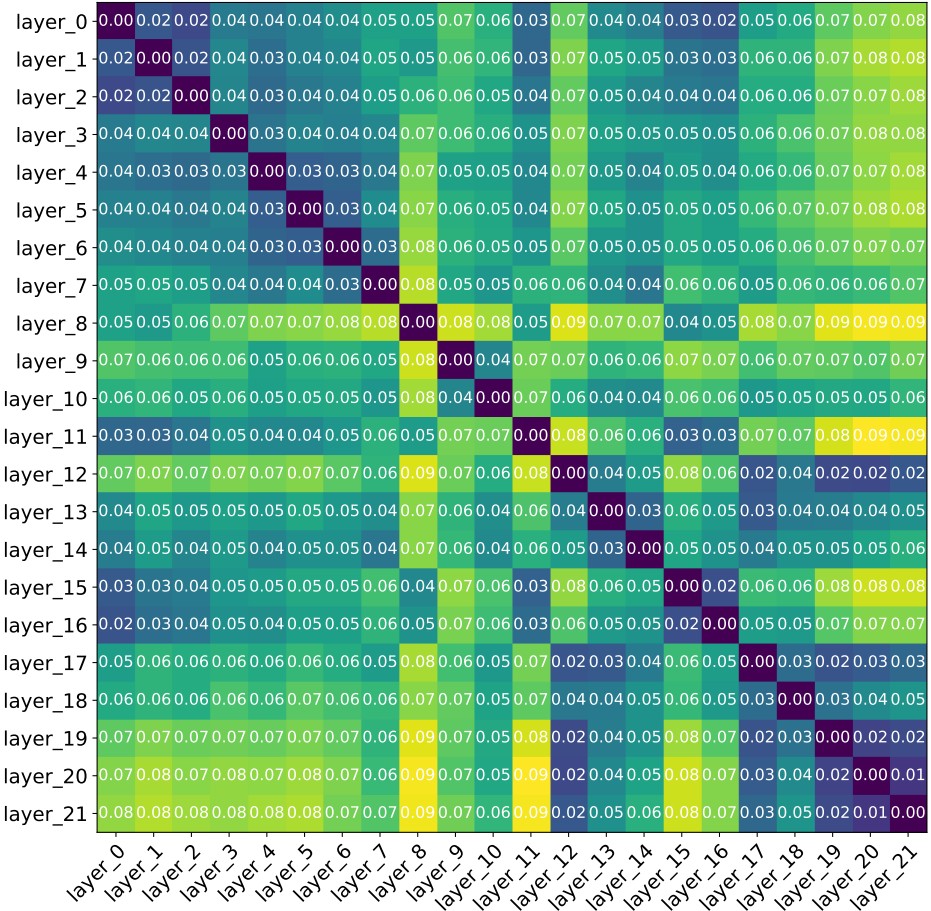

Figure 6: We visualize the disagreement heatmap of $\hat{S}$ selection when the context-dependency score $S_M(\mathbf{Y}_m)$ is calculated across different layers. We select as $\hat{S}$ the 10% of conversation turns with the highest context-dependency scores on each layer. The disagreement is measured by the number of non-overlapping conversation turns in $\hat{S}$ selected by any two layers.

Table 10: Ratio of identified context-dependent instructions in each instruction finetuning dataset. Total number of instructions in each dataset can be found in Table 2

| Dataset | 0.5 | 0.6 | 0.7 | 0.8 |
|---|---|---|---|---|
| ShareGPT (Vicuna) | 0.14 | 0.10 | 0.07 | 0.04 |
| UltraCHat-200K | 0.22 | 0.18 | 0.14 | 0.11 |
| WizardLM-70K | 0.34 | 0.23 | 0.13 | 0.06 |

### B.7 Distribution of Instruction Lengths

Here we visualize the changes in the distribution of instruction lengths between the original instruction finetuning dataset and the selected context-dependent subset $\hat{S}$. Although higher context-dependency is to some extent correlated with longer instruction lengths, there is still a large number of short instructions showing high context dependency and selected for inclusion in $\hat{S}$.

### B.8 Proportion of context-dependent samples filtered with different thresholds

We report the ratio of identified context-dependent instructions with different threshold values $\beta$ in Table 10.

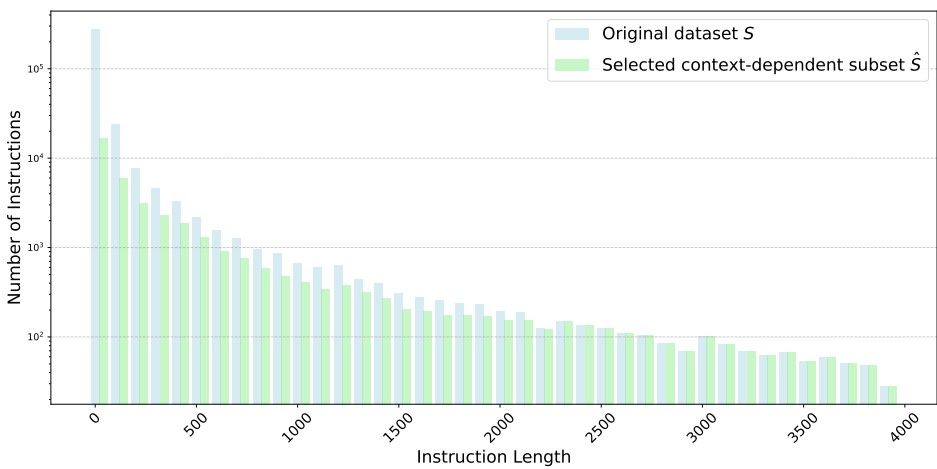

Figure 7: Change of instruction lengths between the original and the selected subset from ShareGPT dataset.

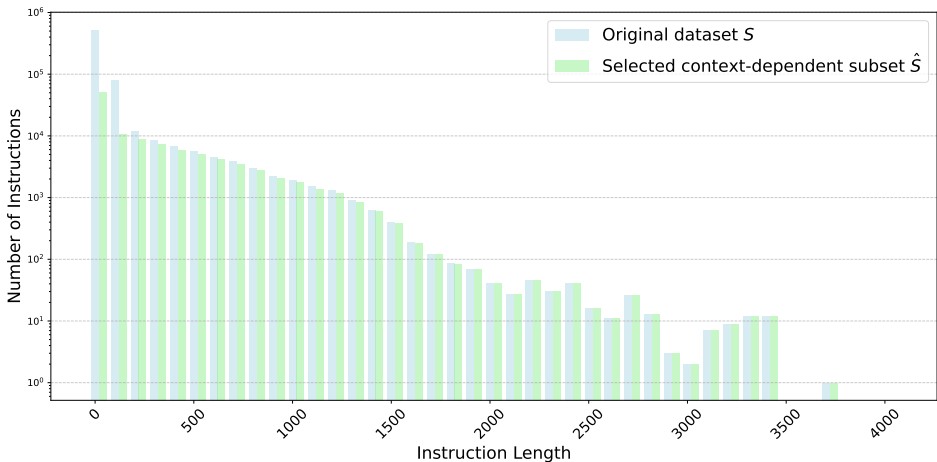

Figure 8: Change of instruction lengths between the original and the selected subset from UltraChat-200K dataset.

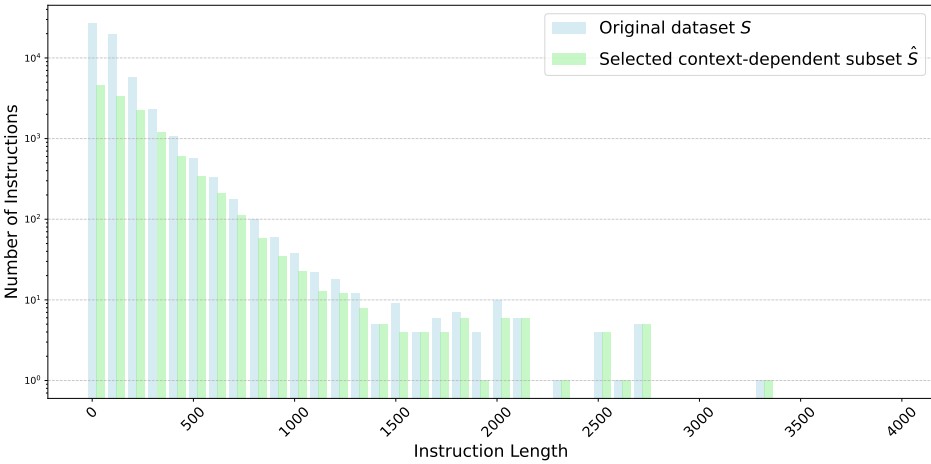

Figure 9: Change of instruction lengths between the original and the selected subset from WizardLM-70K dataset.

