# OpenReview forum: "On the Loss of Context Awareness in General Instruction Fine-tuning"
_NeurIPS.cc/2025/Conference — NeurIPS 2025 poster_

### Official Review · Reviewer_wJSG · 2025-06-28

**Clarity:** 3
**Significance:** 3
**Originality:** 3
**Rating:** 4
**Confidence:** 3

**Summary:**

This paper investigates the degradation of context awareness in LLMs after instruction fine-tuning. The authors show that instruction-tuned models exhibit a bias that reduces attention to user-provided context, particularly when prompted using chat templates. The authors trace this issue to instruction datasets where many responses do not rely on context using the Needle-in-a-Haystack (NIH) task and analysis on attention. They propose to use a token indicating context-dependent instructions during fine-tuning, encouraging the model to reallocate attention appropriately.

**Questions:**

* Can the proposed indicator token method generalize effectively beyond simple retrieval tasks, particularly to tasks requiring complex reasoning or operations?
* Can you analyse further to answer the second point of limitations (above mentioned)?
* Would the proposed method be scalable and effective for larger models or larger datasets?

**Ethical Concerns:**

["NO or VERY MINOR ethics concerns only"]

**Limitations:**

yes

**Paper Formatting Concerns:**

None.

**Quality:**

3

**Strengths And Weaknesses:**

### Strengths
* The paper identifies a specific and underexplored issue in instruction fine-tuning: loss of context awareness due to role-based attention bias introduced by chat templates.
* The analysis of attention on NIH tasks shows how context sensitivity deteriorates after SFT.
* The proposed solution, appending an indicator token to context-dependent instructions, is practical and easy to apply in real-world fine-tuning pipelines.


### Limitations
* The effectiveness of the indicator token approach is clearly shown on the NIH task; however, improvements are marginal or inconsistent on more complex tasks such as DROP and MT-Bench, raising concerns about the generalizability of the proposed method.
* While the paper identifies bias in training data as a root cause, it lacks comprehensive ablation studies. Comparing models fine-tuned on balanced datasets, specifically ones with a higher proportion of context-dependent examples, would provide stronger evidence regarding the influence of data versus model architecture or training procedures.

---

> ### Author Rebuttal · Authors · 2025-07-30
>
> Thank you for your thoughtful review and for recognizing our identification of role-based attention bias and the practicality of our proposed indicator-token solution. Please see the responses to the questions below:
>
> > Q1: Can the proposed indicator token method generalize effectively beyond simple retrieval tasks, particularly to tasks requiring complex reasoning or operations?
>
> A1: Thank you for the suggestion. We conducted an additional evaluation of the fine-tuned Llama-3-8B models on the math_qa dataset, and the resulting accuracies are reported in the table below. Consistent with our findings on DROP, fine-tuning improves the performance of the Wizard and UltraChat models, while it slightly degrades the performance of the ShareGPT models. These results further support our discussion in Section 4.3 that the sparsity of context-dependent instructions can affect performance on complex tasks when using our method.
>
> | | **ShareGPT** | **ShareGPT (ours)** | **UltraChat** | **UltraChat (ours)** | **Wizard** | **Wizard (ours)** |
> | --- | --- | --- | --- | --- | --- | --- |
> | math_qa | $28.87 \pm  0.68$ | $26.19 \pm 0.66$ | $10.73 \pm  0.46$ | $11.66 \pm  0.48$ | $25.77 \pm  0.65$ | $26.7 \pm 0.66$ |
>
> > Q2: Can you analyse further to answer the second point of limitations (above mentioned)?
>
> A2: While we do not have experiments on datasets with an extremely high proportion of context-dependent samples, our work already includes experiments on three fine-tuning datasets that differ in their density of context-dependent examples. As shown in Table 10, the proportion of context-dependent samples increases from ShareGPT to WizardLM. Correspondingly, performance on MT-Bench and DROP improves more significantly on datasets with a higher density of context-dependent conversations (e.g., WizardLM-70K), which also exhibit a more diverse distribution within their context-dependent subset. The additional math_qa results presented above (A1) show a consistent trend. Therefore, although we have not tested datasets where context-dependent samples approach 100%—which is impractical given the broad diversity of use cases—we expect our method’s effectiveness to hold, and potentially improve, as the proportion of context-dependent samples increases.
>
> > Q3: Would the proposed method be scalable and effective for larger models or larger datasets?
>
> A3: Since the issue we identify stems from biases in the post-training data, we expect that larger models would be similarly affected by the same attention bias. Consequently, our method should remain effective for larger models as well. For larger datasets, the impact primarily depends on the data distribution: as long as context-dependent samples do not dominate the dataset, the same bias will persist, and our method should continue to be effective. In other words, the scalability of our approach is determined by the nature of the data rather than the model or data size.

---

> > ### Comment · Reviewer_wJSG · 2025-08-05
> >
> > Thank you for the detailed rebuttal.
> >
> > Q1. I appreciate the inclusion of additional experiments on the math_qa dataset. However, the observed improvements are relatively modest (approximately +1%) and inconsistent across datasets, with performance degradation noted for ShareGPT. As such, these results provide only limited support for the generalizability of the proposed method to more complex reasoning tasks.
> >
> > Q2. The response makes effective use of the existing differences in dataset composition to support the claim regarding context-dependent data density. While this is a reasonable approach, the absence of controlled experiments means the causal impact of data distribution remains somewhat speculative. Nonetheless, I find the rebuttal satisfactory on this point.
> >
> > Q3. The rebuttal on scalability to larger models and datasets is a claim without empirical or theoretical validation. Further experimental evidence would strengthen this aspect of the paper.
> >
> > Overall, I continue to view this submission as a valuable contribution. However, some of the claims would benefit from more thorough validation. I will maintain my score of 4.

---

### Official Review · Reviewer_YvtK · 2025-07-03

**Clarity:** 4
**Significance:** 3
**Originality:** 3
**Rating:** 5
**Confidence:** 4

**Summary:**

This paper uncovers a “role-bias” in instruction-tuned LLMs: when a chat template is used, user tokens receive disproportionately low attention. The authors document the phenomenon on NIH question-answering tasks across Llama, Mistral, and Gemma families. They then propose a mitigation strategy: measure the attention mass on user tokens, and—if it exceeds a tunable threshold—prepend a special marker token to the user instruction during training. Fine-tuning with this marker improves context utilization on NIH, Closed-Book QA, and MT-Bench.

**Questions:**

1. Can the authors provide a comparison experiment to inference time reweighting methods like PASTA? Reweight the attention towards user instructions and see if the proposed addition of special token is better and worth the training efforts. Doesn't need to be large scale experiments but some baselines would significantly strength the work.

2. If best, the authors are encouraged to provide a experiment that scales over 8k tokens given that some major LLM usage scenes nowadays are long context (long COT, multi-turn model environment interactions). The experiments don't need to be on reasoning models or agentic tool call models but some tasks that require longer context will be evidence that the proposed method can generalize to longer inputs.

Q1 : Baseline request
Please include a comparison with attention-reweighting techniques applied only at inference, such as PASTA. An experiment that amplifies attention on user instructions—without retraining—would reveal whether the proposed special-token fine-tuning justifies its additional cost.

Q2: Long-context evaluation
Modern LLM workloads often exceed 8 k tokens (e.g., lengthy chain-of-thought prompts or multi-turn tool-use settings).  The experiments don't need to be on reasoning models or on agentic tool-call tasks but demonstrating the method on a task that truly stresses long contexts would strengthen the claim that it generalizes beyond inputs of moderate-lengths.

**Ethical Concerns:**

["NO or VERY MINOR ethics concerns only"]

**Final Justification:**

This paper uncovers a “role-bias” and proposes a simple yet effective approach to improve user instruction following. I had some misunderstandings about the paper in my original review. But the authors pointed them out and addressed almost all of my comments.
The only remaining experiment I would like to see is how they evaluate on more complex or long-horizon tasks (for example multi-turn, or agentic tasks instead of single turn tasks) but that doesn't diminish the values of this paper. I would recommend acceptance.

**Limitations:**

yes

**Quality:**

4

**Strengths And Weaknesses:**

Strengths
1. Comprehensive empirical evidence supports both the diagnosis of role-bias and the proposed fix. The experiments span three model families, three instruction-tuning corpora, and multiple evaluation suites, showing consistent gains on context-dependent tasks.

2. The manuscript is clear and well-structured, making the core insight and methodology easy to follow.

Weakness
1. Task coverage – NIH and short-span QA test ≤ 8 K tokens; real-world RAG are not studied.
It would be beneficial to add one or two tasks that require long context (>8k) to showcase the proposed solutions can scale up to longer context.

2. Dependency on a seed model & threshold:
Context-dependency score depends on head selection and a heuristic β. In practition, if we want to scale up training and include more datasets, I can imagine that this threshold needs to be tuned.

---

> ### Author Rebuttal · Authors · 2025-07-30
>
> Thank you for your detailed review and for appreciating the clarity of our manuscript as well as the breadth and consistency of our empirical evidence across models, datasets, and evaluation suites. Please see the responses to the questions below:
>
> > Q1: Baseline request Please include a comparison with attention-reweighting techniques applied only at inference, such as PASTA. An experiment that amplifies attention on user instructions—without retraining—would reveal whether the proposed special-token fine-tuning justifies its additional cost.
>
> A1: We believe there may be a misunderstanding, as Section 2.3 of the paper is entirely dedicated to inference-time attention steering methods (specifically PASTA). To reiterate our findings, such attention steering can partially mitigate the reduced context-awareness. However, stronger steering magnitudes introduce negative side effects, which limit its effectiveness compared to the fine-tuning approach (see Page 5 Line 171).
>
> > Q2: Long-context evaluation Modern LLM workloads often exceed 8 k tokens (e.g., lengthy chain-of-thought prompts or multi-turn tool-use settings). The experiments don't need to be on reasoning models or on agentic tool-call tasks but demonstrating the method on a task that truly stresses long contexts would strengthen the claim that it generalizes beyond inputs of moderate-lengths.
>
> A2: We agree with the reviewer that it is important to verify whether the identified issue also appears in long-context settings. As shown in Fig. 2 (Table 8), our NIH benchmark does evaluate models at their full context lengths, which range from 4K up to 128K tokens (e.g., 32K for Mistral models and 128K for Llama‑3.1). Due to computational constraints, we were unable to fine-tune models beyond 8K context for additional complex tasks. Nevertheless, since the observed reduction in context-awareness persists even in models with extended context windows, we expect our proposed method to remain effective in these long-context scenarios.
>
> > W2: Context-dependency score depends on head selection and a heuristic β. In practition, if we want to scale up training and include more datasets, I can imagine that this threshold needs to be tuned.
>
> We appreciate the reviewer’s concern regarding the dependence on hyperparameters when applying our method to new settings. As described on Page 6 Line 207, the attention head is selected by identifying the head that assigns the highest weight to user tokens. Importantly, the seed model used for head selection does not need to match the fine-tuned model, as demonstrated by our experiments. For the heuristic choice of $\beta$, we provide a sensitivity analysis in Appendix B.5, which shows that the performance of our method is robust and not overly sensitive to the specific value of this threshold.

---

> > ### Comment · Reviewer_YvtK · 2025-08-04
> >
> > It's my mistake to not identify the attention steering you used is PASTA. Comparing to this baseline makes the paper contribution stronger.
> > Also thanks for addressing my concerns about token number scaling and head/threshold selection. I agree training on long context requires massive compute being a challenge.

---

### Official Review · Reviewer_uYwu · 2025-07-03

**Clarity:** 4
**Significance:** 3
**Originality:** 4
**Rating:** 5
**Confidence:** 4

**Summary:**

The author identifies the loss of context-awarness measured by NIH test after fine-tuning a pre-trained model, and shows that the observation is related to the use of chat template. The author further shows evidence that the root cause can be the shift of attention weight toward assistant tokens when chat template is applied during SFT. Based on the observation, the author proposes a fine-tuning techniques that identifies and adds a context-awareness indicator to the fine-tuning data, and shows general improvement after applying the technique on various combinations of models and tasks.

**Questions:**

1. It would be interesting to include results when different models (potentially the original pre-trained model) are used to determine the context-aware indicator. The results would give further insights on how the capabilities of indicator model affects the recovery of context-awareness.

**Ethical Concerns:**

["NO or VERY MINOR ethics concerns only"]

**Limitations:**

Yes.

**Paper Formatting Concerns:**

The reviewer does not have major formatting concerns.

**Quality:**

4

**Strengths And Weaknesses:**

Strength:
1. The paper is very well-written and well-motivated.
2. The paper presents concise and convincing study of the loss of context awareness and its root cause.
3. The method developed based on manual modification is simple yet effective.

Weakness:
1. It seems that the loss of context-awareness is not catastrophic. Therefore, it is unclear to the reviewer whether paying the compute cost to generate context-aware indicators is worthwhile when the user perform SFT in general scenarios.

---

> ### Author Rebuttal · Authors · 2025-07-31
>
> Thank you for your thoughtful review and for recognizing the clarity, motivation, and effectiveness of our study and proposed method. Please see the response to the question below:
>
> > Q1: It would be interesting to include results when different models (potentially the original pre-trained model) are used to determine the context-aware indicator. The results would give further insights on how the capabilities of indicator model affects the recovery of context-awareness.
>
> A1: We thank the reviewer for pointing out this aspect, as it highlights an important benefit of our approach. Our current implementation uses a different model to compute the context-aware indicator than the one being fine-tuned. Specifically, the indicator is generated using a TinyLlama model fine-tuned on the ShareGPT data (Page 7, Line 261). This indicator is then reused across all experiments with different pretrained models. This design addresses the reviewer’s concern in the weakness section: the indicator only needs to be computed once per dataset, and the model used for this computation can be significantly smaller, minimizing the computational cost.

---

> > ### Comment · Reviewer_uYwu · 2025-08-05
> > **Thanks for the rebuttal**
> >
> > It is indeed a good take that reusable indicator generators minimize the overall overhead across different models, and I appreciate the take. I still would like to see how, for example, using the original pretrained model would improve the performance further or not, as one would expect that indicator model choices tailored for each model would be more "accurate". However that is not mandatory from my point of view.

---

### Official Review · Reviewer_3Mu4 · 2025-07-17

**Clarity:** 4
**Significance:** 4
**Originality:** 4
**Rating:** 5
**Confidence:** 4

**Summary:**

The paper shows that instruction tuning, particularly with chat templates, weakens the model’s ability to remain context-aware. The paper further discusses that during conversational finetuning, models become biased toward specific roles, as seen through attention patterns. It was found that user tokens get less attention with templates. While steering more attention to user tokens improves context awareness (like on the NIH dataset), it hurts performance on others like DROP.  To address this, the paper introduces a new conditioning method that measures how much a model’s response depends on earlier user instructions. This context dependency is indicated by a score which is generated via reference LLM. If a conversation turn is context-dependent, a special token [IND] is added to the user instruction. This approach helps the model perform better on both context-aware and complex tasks.

**Questions:**

None

**Ethical Concerns:**

["NO or VERY MINOR ethics concerns only"]

**Limitations:**

Yes

**Quality:**

3

**Strengths And Weaknesses:**

Strength

 - I liked the idea and flow in which the methods are derived in a coherent, step-by-step manner that makes the paper easy to follow.
- The paper is well-written, making the proposed method accessible to readers
- The paper uses a solid and relevant set of models to support its findings.
- Proposes two ways to mitigate context awareness in LLMs. The claims are supported with empirical evidence for most part of study.


Weaknesses

- The paper uses only the NIH dataset to show the negative impact of chat templates. Evaluating at least top 1 or 2 models on one more context aware dataset will strengthen the claim. It would also help if performance on complex skills is not added to understand how models perform on these before and after instruction tuning.

- The attention steering approach appears limited to a specific type of attention mechanism. A discussion on how the method could be extended to other attention implementations would improve understanding of its generalizability.

- The paper would benefit from a direct comparison, ideally with graphs or figures, showing how attention to different prompt components differs between the two methods: attention steering and conditional instruction fine-tuning.

---

> ### Author Rebuttal · Authors · 2025-07-31
>
> Thank you for your thoughtful review and for appreciating the clarity of our presentation, the relevance of our model choices, and the coherence of our proposed methods. Please see the responses to the mentioned weaknesses below:
>
> > W1: The paper uses only the NIH dataset to show the negative impact of chat templates. Evaluating at least top 1 or 2 models on one more context aware dataset will strengthen the claim. It would also help if performance on complex skills is not added to understand how models perform on these before and after instruction tuning.
>
> A1: In response to the reviewer’s suggestion, we explored simpler QA-style tasks to isolate the impact of chat templates. We found that the pretrained (non-instruction-tuned) models were unable to answer even simple questions, despite adding hints to the prompts. Nevertheless, we conducted a comparison between instruction-tuned models with and without chat templates on LongBench, a benchmark specifically designed to evaluate context awareness with relatively straightforward questions that do not require complex reasoning. Performance was measured using a bag-of-words recall metric, and the results are as follows:
>
> | | Llama-3-8B-Instruct | Llama-3.1-8B-Instruct |
> | --- | --- | --- |
> | w/ chat template | 0.3504 |0.5176 |
> | w/o chat template | **0.3662** | **0.5267** |
>
> We observe that removing the chat template consistently improves performance, which further supports our claim that the use of role tokens introduces a bias that reduces context awareness.
>
> > W2: The attention steering approach appears limited to a specific type of attention mechanism. A discussion on how the method could be extended to other attention implementations would improve understanding of its generalizability.
>
> A2: While our attention steering experiments were conducted using standard scaled dot-product attention, the method is not inherently tied to this mechanism. In the revision, we will add a discussion section describing how the same idea could be adapted to architectures with alternative attention mechanisms (e.g., multi-query or linear attention) since it only relies on reweighting cross-token dependencies.
>
> > W3: The paper would benefit from a direct comparison, ideally with graphs or figures, showing how attention to different prompt components differs between the two methods: attention steering and conditional instruction fine-tuning.
>
> A3: We thank the reviewer for suggesting a more direct comparison between attention steering and conditional fine-tuning. We will add a new figure to the supplementary material that visualizes the attention distribution differences between these two methods across prompt components. The attention scores of the user (assistant) tokens for the steering are merely multiplied (divided) by a constant factor. This further highlights why conditional fine-tuning provides a more flexible trade-off across tasks.

---

> > ### Comment · Reviewer_3Mu4 · 2025-08-06
> > **Thank you for the responses.**
> >
> > Thank you for clarifying the points and adding new results. I hope the authors add the graphs as mentioned in the response.

---

### Comment · Area_Chair_9pt9 · 2025-08-01

Dear Reviewers,

The authors have posted their rebuttals. If you have any additional questions or require clarification, please add your comments as soon as possible—the author-reviewer discussion period ends in one week.

Thank you for your prompt attention.

Best regards,

AC

---

> ### Comment · Area_Chair_9pt9 · 2025-08-06
>
> Here is a gentle reminder: The Author-Reviewer discussions will end by Aug 8, 11.59pm AoE. Please also be aware that there is a "Mandatory Acknowledgement" for the reviewers.

---

### Decision · Program_Chairs · 2025-09-17

**Decision:**

Accept (poster)

**Comment:**

The paper investigates how instruction fine-tuning with chat templates reduces LLMs’ context awareness and identifies role-bias in attention patterns as a root cause. The authors propose conditional fine-tuning with a context-dependency indicator token, demonstrating consistent improvements across multiple models and tasks without sacrificing general instruction-following ability. Strengths highlighted by reviewers include the clarity of exposition, thorough empirical analysis across datasets and model families, and a simple yet effective mitigation strategy. Weaknesses noted were limited evaluations on long-context and complex reasoning tasks, reliance on a heuristic threshold, and modest gains in some cases. However, during rebuttal the authors addressed these concerns with additional experiments (e.g., LongBench, math_qa) and clarifications on generalizability, scalability, and computational cost, which reviewers acknowledged as satisfactory even if some open questions remain. Overall, the contribution is original, timely, and practically relevant, making a compelling case for acceptance.